

# Development of a High-Resolution Integrated Emission Inventory of Air Pollutants for China

Nana Wu[1], Guannan Geng[2,3*], Ruochong Xu[1], Shigan Liu[1], Xiaodong Liu[2], Qinren Shi[2], Ying Zhou[4], Yu Zhao[5], Huan Liu[2,3], Yu Song[6], Junyu Zheng[7], and Qiang Zhang[1]

[1]Ministry of Education Key Laboratory for Earth System Modeling, Department of Earth System Science, Tsinghua University, Beijing 100084, China
[2]State Key Joint Laboratory of Environment Simulation and Pollution Control, School of Environment, Tsinghua University, Beijing 100084, China
[3]State Environmental Protection Key Laboratory of Sources and Control of Air Pollution Complex, Beijing 100084, China
[4]Key Laboratory of Beijing on Regional Air Pollution Control, Faculty of Environment and Life, Beijing University of Technology, Beijing, 100124, China
[5]State Key Laboratory of Pollution Control and Resource Reuse and School of the Environment, Nanjing University, 163 Xianlin Rd., Nanjing, Jiangsu 210023, China
[6]State Key Joint Laboratory of Environmental Simulation and Pollution Control, College of Environmental Sciences and Engineering, Peking University, Beijing 100871, PR China
[7]Sustainable Energy and Environmental Thrust, The Hong Kong University of Science and Technology (Guangzhou), Guangzhou, 511458, China

*Correspondence to*: Guannan Geng (guannangeng@tsinghua.edu.cn)

**Abstract.** Constructing a highly-resolved comprehensive emission dataset for China is challenging due to limited availability of refined information for parameters in a unified bottom-up framework. Here, by developing an integrated modeling framework, we harmonized multi-source heterogeneous data including several up-to-date emission inventories at national and regional scale, and for key species and sources in China, to generate a 0.1 °resolution inventory for 2017. By source mapping, species mapping, temporal disaggregation, spatial allocation and spatial-temporal coupling, different emission inventories are normalized in terms of source categories, chemical species, and spatiotemporal resolutions. This achieves the coupling of multi-scale, high-resolution emission inventories with the MEIC (Multi-resolution Emission Inventory for China), forming a high-resolution INTegrated emission inventory of Air pollutants for China (i.e., INTAC). We find that the INTAC provides more accurate representations for emission magnitudes and spatiotemporal patterns. In 2017, China's emissions of $SO_2$, $NO_x$, CO, NMVOC, $NH_3$, $PM_{10}$, $PM_{2.5}$, BC, and OC are 12.3, 24.5, 141.0, 27.9, 9.2, 11.1, 8.4, 1.3 and 2.2 Tg, respectively. The proportion of point source emissions for $SO_2$, $PM_{10}$, $NO_x$, $PM_{2.5}$ increases from 7–19% in MEIC to 48–66% in INTAC, resulting in improved spatial accuracy, especially mitigating overestimations in densely populated areas. Compared to MEIC, INTAC reduced mean biases in simulated concentrations of major air pollutants by 2–14 µg/m³ across 74 cities against ground observations. The enhanced model performance by INTAC was particularly evident at finer grid resolutions. Our new dataset is accessible at https://doi.org/10.5281/zenodo.10459198 (Wu et al., 2024), and it will provide a solid data foundation for fine-scale atmospheric research and air quality improvement.



## 1 Introduction

In recent years, China has achieved remarkable progress in improving air quality and public health through the active implementation of clean air policies (Xiao et al., 2022; Zhang et al., 2019a; Liu et al., 2020; Zhang and Geng, 2019). To further unlock the potential of targeted clean air actions, there is an urgent need for an accurate and detailed depiction for emissions, encompassing their magnitudes and spatial-temporal patterns. Developing a reliable highly-resolved emission inventory for China is also crucial for studies of atmospheric chemistry and climate change (Zhang et al., 2019a; Geng et al., 2021; Cheng et al., 2021a).

The construction of high-resolution emission inventories for China poses significant challenges due to the diversity and complexity of emission sources and technology distributions. Additionally, the limited availability of localized measurements for emission factors (EFs) and source profiles, along with exact location of the emission facilities, further compounds the difficulties (Li et al., 2017a). The widely-used bottom-up approach involves the establishment of a unified framework that encompasses source categories, chemical speciation processes, spatial-temporal allocation profiles and emission estimation methods (Huang et al., 2021; An et al., 2021). However, achieving both wide coverage and high accuracy in compiling an emission inventory for China through this approach remains a formidable task for individual research institutions.

Comprehensive national-scale emission inventories developed using the unified framework typically provide extensive coverage of space, species and sectors (Li et al., 2017a; Li et al., 2023b), but tend to exhibit limitations in spatial accuracy (Wu et al., 2021; Zhou et al., 2017a; Zhao et al., 2015; Zheng et al., 2021). Previous studies have indicated that the spatial allocation in large-scale emission inventories rely on spatial proxies (e.g., population, road networks) rather than latitude-longitude coordinates of emission sources due to the unavailability of extensive spatial information (Zhang et al., 2009; Li et al., 2017b). The assumption of a linear correlation between emissions and spatial proxies might lead to an overestimation of emissions in urban areas, especially at scales finer than 0.25 ° (Zheng et al., 2017; Zheng et al., 2021; Wu et al., 2021). Biases introduced by the proxy-based method are found to be propagated as the grid size diminishes, resulting in uncertainties for chemical transport models (CTMs) (Zheng et al., 2017; Zheng et al., 2021).

Emission inventories focused on a specific region (Liu et al., 2018; An et al., 2021; Huang et al., 2021), sector (Deng et al., 2020; Chen et al., 2016; Zhou et al., 2017b) or key species (Li et al., 2021; Huang et al., 2012b; Wang et al., 2023) under the aforementioned unified framework demonstrate enhanced accuracy, but fail to achieve comprehensive coverage. These inventories assimilate substantial detailed foundational data from various statistical dataset, on-site measurements or surveys to represent real-world emission magnitudes, including energy consumption, removal efficiencies, and localized speciation profile (Liu et al., 2018; An et al., 2021; Huang et al., 2021). Innovative data, such as measurements from continuous emission monitoring systems (Tang et al., 2023; Bo et al., 2021; Wu et al., 2022), or methodologies like process-based models (Kang et al., 2016; Zhao et al., 2020) are implemented to enable a more accurate characterization of complex emission dynamics. Facility-level geographic location is incorporated to optimize the representation of spatial patterns (Liu et al., 2015a; Wu et



al., 2023; Wang et al., 2019). The reliability of these local-scale, sector- or species-specified inventories has been validated
against satellite and ground-based measurements (Zhang et al., 2021; Liu et al., 2016a; Zheng et al., 2019).
The other strategy for developing bottom-up emission inventories is commonly known as the integrated method. This method
consolidates emission datasets for various regions, species or sectors into a unified product, ensuring extensive coverage (Li
et al., 2017b). Taking advantage of existing inventories derived from localized data and advanced methods, the integrated
method facilitates the efficient generation of highly-resolved emission inventories at large scales. However, the heterogeneity
of different emission datasets presents challenges for the fusion, manifested in diverse data formats, sector categories, species,
spatial-temporal resolution. In recent years, there has been growing interest in adopting the integrated approach to enrich
inventories with local insights, particularly at the global  (Janssens-Maenhout et al., 2015; Crippa et al., 2023) and Asian scales
(Zhang et al., 2009; Kurokawa et al., 2013; Li et al., 2023a; Li et al., 2017b). Researches on establishing integrated inventories
for China are constrained due to the inherent complexity and challenging accessibility of the data. These efforts are
concentrated in specific regions, such as the Yangtze River Delta (An et al., 2021).
In this work, with the support of several research institutions, we use an emission integration model to construct a high-
resolution integrated emission inventory at a spatial resolution of 0.1 ° for China in 2017, denoted as INTAC. The challenges
associated with coupling multi-source heterogeneous data are addressed through the implementation of an inventory
integration framework. Then, leveraging the strengths of inventories enriched with local knowledge, we compile a
comprehensive highly resolved emission product to enhance the accurate representation of emissions from crucial regions,
sectors and species. Finally, the improved accuracy of emission magnitude and spatial distribution is evaluated using
atmospheric chemistry models.
**2 Methodology and data**
Figure 1 illustrates the schematic diagram of the integration process of INTAC. We collect seven emission inventories—MEIC
developed by Tsinghua University (Li et al., 2017a; Zheng et al., 2018), the industrial point source emission inventory for
China by the MEIC team (Zheng et al., 2017; Zheng et al., 2021), the Yangtze River Delta (YRD) air pollutant emission
inventory led by Nanjing University (An et al., 2021; Zhou et al., 2017a), the Pearl River Delta (PRD) emission inventory by
Jinan University (Huang et al., 2021; Sha et al., 2021), the open biomass burning emission inventory in China by Peking
University (Song et al., 2009; Huang et al., 2012a; Yin et al., 2019; Liu et al., 2015b) , the shipping emission inventory in East
Asia by Tsinghua University (Liu et al., 2016b; Liu et al., 2019) , and the high-resolution ammonia emission inventory in
China (PKU-NH$_3$) by Peking University (Huang et al., 2012b; Kang et al., 2016). The details of these inventories and the
rationale for choosing them will be described in Sect. 2.1.



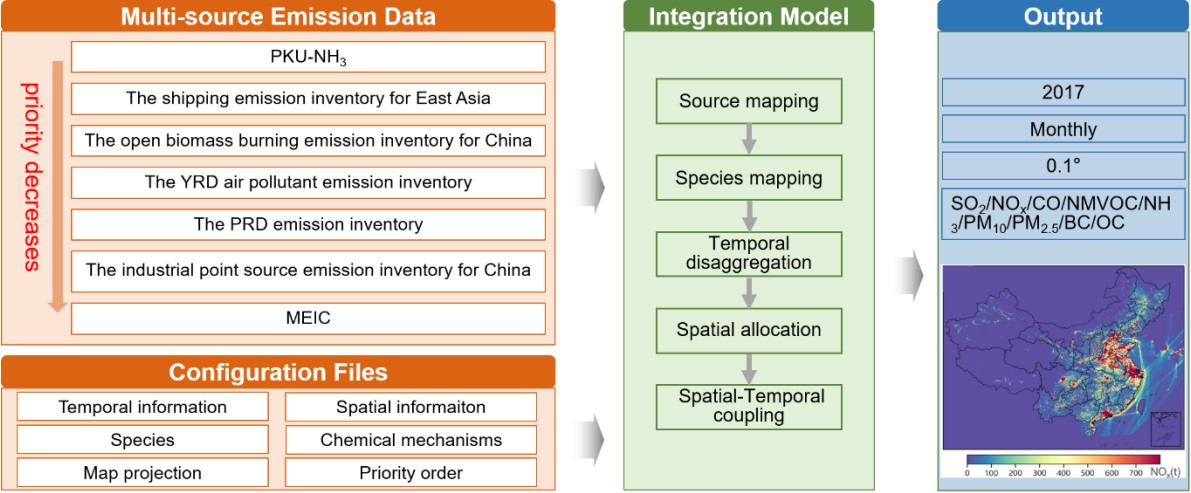

**Figure 1: Methodology framework of the INTAC inventory development.**
An integration model is then established to merge together emission inventories with different sectors, species, spatial-temporal
resolution and formats (i.e., point, area, and gridded forms). The integration process consists of five steps: source mapping,
species mapping, temporal disaggregation, spatial allocation, and spatial-temporal coupling, as detailed in Sect. 2.2. Based on
the priority order, multi-source emission inventories are assembled at the standardized specie, sector, and grid levels, yielding
a standardized data cube. Ultimately, an integrated emission inventory is created for China, featuring a resolution of $0.1°$ on a
monthly scale and covering nine air pollutants (i.e., $SO_2$, $NO_x$, $CO$, $NMVOC$, $NH_3$, $PM_{10}$, $PM_{2.5}$, $BC$, $OC$).
**2.1 Components of the integrated emission inventory INTAC**
Table 1 lists the essential details about the seven inventories and priority order utilized for integration. As the most widely
used anthropogenic emission inventory in China, MEIC functions as the default inventory in our integration, supplementing
missing data in other inventories. The remaining six emission inventories, demonstrating proficiency in specific species,
sectors, or regions, are coupled with MEIC to improve the representation of national emissions.





**Table 1: List of emission inventories collected in this work.**

| Priority ranking | Emission inventory and developer | Year | Resolution | Region | Resolution | Species |
|---|---|---|---|---|---|---|
| 1 | PKU-NH$_3$ (Peking University) | 1980–2017 | Monthly | Mainland China | 0.1° | NH$_3$ |
| 2 | The shipping emission inventory for East Asia (Tsinghua University) | 2017 | Annually | East Asia | 0.1° | SO$_2$/NO$_x$/CO/NMVOC/ PM$_{2.5}$/BC/OC |
| 3 | The open biomass burning emission inventory for China (Peking University) | 1980–2017 | Daily | Mainland China | ~1km | SO$_2$/NO$_x$/CO/NMVOC/ NH$_3$/PM$_{10}$/PM$_{2.5}$/BC/OC |
| 4 | The PRD emission inventory (Jinan University) | 2017 | Monthly | PRD | 0.05° | SO$_2$/NO$_x$/CO/NMVOC/ NH$_3$/PM$_{10}$/PM$_{2.5}$/BC/OC |
| 5 | The YRD emission inventory (Nanjing University/Shanghai Academy of Environmental Sciences/Jiangsu Provincial Academy of Environmental Science) | 2017 | Annually | YRD | 0.1° | SO$_2$/NO$_x$/CO/NMVOC/ NH$_3$/PM$_{10}$/PM$_{2.5}$/BC/OC |
| 6 | The industrial point source emission inventory for China (Tsinghua University) | 2012–2018 | Monthly | Mainland China | ~1km | SO$_2$/NO$_x$/CO/NMVOC/ NH$_3$/PM$_{10}$/PM$_{2.5}$/BC/OC |
| 7 | MEICv1.3 (Tsinghua University) | 2008–2017 | Monthly | Mainland China | 0.25° | SO$_2$/NO$_x$/CO/NMVOC/ NH$_3$/PM$_{10}$/PM$_{2.5}$/BC/OC |

**2.1.1 MEIC**
The integrated inventory INTAC is built upon MEIC, a comprehensive database with extensive coverage across time periods,
space, species, and sectors. Developed by Tsinghua University since 2010 (http://meicmodel.org.cn) (Li et al., 2017a; Zheng
et al., 2018), the MEIC provides monthly emissions for air pollutants and CO$_2$ in China from 1990 to the present at a resolution
of 0.25° × 0.25°. It caters to the demand for timely and accurate estimates of atmospheric emissions, gaining widespread
adoption by both domestic and international research institutions. We use 2017 emissions from MEIC v1.3 in this study.
MEIC employs several strategies to improve emission estimation parameters. This includes categorizing emission sources
across ~800 sectors, utilizing a technology- and big-data-driven approach for dynamic emission characterization, and



employing a localized emission factor database (Li et al., 2017a; Zheng et al., 2018). Emission estimates for power, on-road,
and residential sources are enhanced through the use of unit-level data (Liu et al., 2015a), county-level emission estimates
(Zheng et al., 2014), and integration of extensive household surveys (Peng et al., 2019), respectively. MEIC builds an database
encompassing temporal allocation profiles (ranging from yearly to monthly, daily, and hourly) (Li et al., 2017b), spatial
allocation proxies (from province to county, and further to grids) (Li et al., 2017b; Geng et al., 2017; Zheng et al., 2017), and
a speciation framework for NMVOC involving five mechanisms (CB-IV, CB05, SAPRC-07, SAPRC-99, and RADM2) (Li et
al., 2014) (Li et al., 2014) to support the development of model-ready gridded emissions.
Among the seven inventories, MEIC has the lowest priority, and is only considered when the other six cannot provide necessary
data for emissions from a specific city and source.

### 2.1.2 The industrial point source emission inventory for China

The proxy-based method used for spatial allocation in MEIC introduces biases in emission mapping, especially at kilometer
scale (Zheng et al., 2017; Zheng et al., 2021). To significantly reduce the uncertainty, we merged an industrial emission
inventory with detailed information on ~100,000 facilities into INTAC.
Compiled by Tsinghua University for the year 2013 (Zheng et al., 2021) and updated by the same research team for 2017, this
point-based inventory combines three databases investigated under the guidance of the Chinese government, offering a
comprehensive overview of industrial facilities. It includes details on locations, activity rates, production technology, end-of-
pipe pollution control devices, and other parameters. The facility-level, technology-based approach allows for dynamic
tracking of emission fluctuations resulting from technological advancements and tightening emission regulations. Crucially,
the use of facility geolocations rather than relying on spatial proxies like urban population enables the derivation of gridded
industrial data at a resolution of ~1 km. This approach significantly avoids misallocating emissions from rural to urban areas
at fine grids, as supported by previous studies demonstrating its effectiveness in mitigating simulated biases in air pollutant
concentrations within densely populated regions (Zheng et al., 2021). For temporal variations, it employs the same monthly
profiles as MEIC, including the production of various industrial goods or GDP, as outlined in Li et al. (2017b). The NMVOC
speciation also aligns with the MEIC model. This inventory takes priority in integration over MEIC, indicating that only few
industrial sources not covered in this inventory are substituted with MEIC.

### 2.1.3 The YRD air pollutant emission inventory

Regional emission inventories in YRD provide a more accurate representation of emissions compared to the national-scale
MEIC, as proven by ground and satellite observations (Zhou et al., 2017a; Zhao et al., 2018; Zhao et al., 2020; Zhang et al.,
2021; Zhao et al., 2017b; Yang and Zhao, 2019). This improvement is attributed to the avoidance of outdated or non-localized
emission calculation parameters, commonly present in large-scale inventories like MEIC. Here, we merge the 2017 YRD air
pollutant emission inventory into INTAC for state-of-the-art estimates of rapidly changing emissions over this core area (An
et al., 2021; Gu et al., 2023; Zhou et al., 2017a).





Localized field surveys and measurements greatly enhance the reliability of calculation parameters within the YRD inventory.
Highly-resolved emissions for the power sector are acquired through on-site monitoring with high temporal resolution (Zhang
et al., 2019b), rather than relying on static and outdated average emission factors. Facility-level information (e.g., the removal
efficiencies) obtained from local investigation and a segment-based industrial process method enhances understanding of both
the quantity and spatial patterns of industrial emissions. Considering meteorological factors and land use conditions during
agricultural processes results in more accurate seasonal and spatial distributions of $NH_3$ emissions. (Zhao et al., 2020). An
investigation of in-use machinery is conducted to capture the seasonal emission patterns from off-road machines (Zhang et al.,
2020). Real-world surveys are performed to determine grain straw ratios and household burning proportions, facilitating the
quantification of emissions from biomass-fueled stoves. The $PM_{2.5}$ and NMVOC speciation profiles are updated based on
multi-instrument sampling and analysis in both current and previous studies (Huang et al., 2018; Zhao et al., 2017b), satisfying
the needs for simulating $PM_{2.5}$ chemical components and $O_3$. The YRD inventory is collected with a spatial resolution of 0.1
degree and an annually temporal resolution in this study. Only CB05 VOC species are collected.

**161  2.1.4 The PRD emission inventory**

The regional emission inventories in the RPD region demonstrated enhanced reliability compared to previous studies (Huang
et al., 2021; Sha et al., 2021; Zheng et al., 2012). The PRD emission inventory in this study captures spatial and temporal
variations within the Pearl River Delta region under emission control policies, serving as a foundation for supporting air quality
modeling (Huang et al., 2021; Sha et al., 2021).
The PRD inventory exhibits notable accuracy improvements, achieved through big data-driven estimation methods, updated
spatial-temporal allocations, and localized NMVOC speciation profiles. Gridded hourly open biomass burning emissions are
quantified by fusing the fire radiative power data from three satellites, and hourly shipping emissions are estimated using high-
frequency AIS records. Thirty-one monthly profiles and ten spatial proxies are updated to reflect spatial-temporal patterns of
emissions influenced by economic growth and energy consumption adjustment. Approximately 90% of industrial emissions
are disaggregated using exact locations, and novel proxies (e.g., farmland production potential) have been developed for
several sectors. The NMVOC speciation is carried out through massive localized measurements and literature reviews,
manifested as a collection of 480 NMVOC source profiles across eight sectors and 380 species. The species relevant to the
SAPRC-07 chemical mechanism are collected here. Additionally, the inventory encompasses 800 source categories, placing
particular emphasis on incorporating new sectors relevant to VOC emissions. Activity rates are improved through extensive
field surveys and data mining efforts, involving investigations of production data for 10,000 industrial plants and the gathering
of activity-relevant information for 50 million vehicles. Emission factors that reflect local context are obtained or revised based
on source measurements and latest research findings. These updates help mitigate uncertainties in emission estimates for the
PRD region. The PRD inventory is initially presented at a monthly resolution and a spatial resolution of 0.05 °, with detailed
spatial-temporal allocation proxies outlined in Huang et al. (2021).





### 2.1.5 The open biomass burning emission inventory in China

As a significant source of $CO_2$, BC, OC and other pollutants, open biomass burning profoundly influences air quality, climate change, and human health (Reisen et al., 2013). A case study in summer 2011 for the YRD region revealed that during a severe haze episode, open biomass burning contributed to 37%, 70%, and 61% of $PM_{2.5}$, OC, and EC emissions, respectively (Cheng et al., 2014). To address the absence of this source in MEIC, we integrate a high-resolution open biomass burning emission inventory from Peking University into INTAC (Song et al., 2009; Huang et al., 2012a; Yin et al., 2019; Liu et al., 2015b). The inventory applies satellite observations to tackle considerable uncertainties tied to provincial statistical data and overcome the coarse resolution found in previous studies (Ni et al., 2015). The estimation of biomass consumption in the inventory is based on the fire radiative energy (FRE) approach, which depends on the energy emitted by fires. This approach helps reduce the biases introduced by burned areas algorithms, especially for small-scale fires. The inventory utilizes the high spatial resolution land cover dataset GlobeLand30 derived from multispectral images to classify biomass fuel types. Eventually, daily emissions from forest, grassland, cropland and shrubland are calculated at a 1-kilometer resolution. The reasonableness is validated by comparing with other datasets, such as the fourth version of the Global Fire Emissions Database. The initially collected inventory lacks VOC model species.

### 2.1.6 The shipping emission inventory in East Asia

In recent years, maritime trade in the East Asian region has significantly increased (Trade and Development, 2014), resulting in a surge in shipping emissions with substantial impacts on air quality and climate. Previous studies have indicated that East Asian shipping emissions accounted for 16% of the global total in 2013. Shipping emissions made a growing contribution to the rise in annual mean $PM_{2.5}$ concentrations, reaching levels as high as 5.2 μg/m$^3$ in 2015 (Lv et al., 2018). To address the omission of this emission source in the MEIC, we integrate the shipping emission inventory in East Asian for 2017 into INTAC (Liu et al., 2016b; Liu et al., 2019).

The inventory introduces an innovative approach based on comprehensive and dynamic ship activity data. A static dataset of approximately 66,000 vessels is compiled as a foundation, using information from Lloyd's Register and China Classification Society. This dataset encompasses various ship properties, including ship category, hull shape, engine rotational speed, engine capacity, maximum speed capability, build year, and more. High quality Automatic Identification System (AIS) data is used to capture ship activities, incorporating the Maritime Mobile Service Identification identifier, geographical location, real-time speed, and time-related information. This data is also employed to generate gridded emissions from shipping at a spatial resolution of 0.1 °. The inventory enhances our comprehension of regional-level shipping emissions and significantly alleviates biases arising from the misallocation of marine fuels, as observed in global studies (Endresen et al., 2007). The collected shipping inventory provides emissions at an annually resolution for seven species, including $SO_2$, $NO_x$, CO, NMVOC, $PM_{2.5}$, BC, and OC.



### 2.1.7 PKU-NH₃

As a prominent alkaline component in the atmosphere, ammonia plays a crucial role in atmospheric chemistry, terrestrial and aquatic ecosystems through its participation in atmospheric reactions and deposition processes. This study integrates PKU-$NH_3$, a high-resolution ammonia emission inventory for China developed by Peking University. PKU-$NH_3$ is designed to track the evolution of $NH_3$ emissions amid the rapid increase in grain and meat production in China over the past few decades (Huang et al., 2012b; Kang et al., 2016). This inventory offers a better grasp on $NH_3$ emissions in China through the application of a processed-based method and more reliable emission factors, in contrast to previous studies (Kurokawa et al., 2013; Li et al., 2017b).

Earlier studies of $NH_3$ emissions commonly used fixed EFs, overlooked some ammonia emission sources, and had coarse resolutions (Streets et al., 2003; Ohara et al., 2007). Unlike previous approaches, the PKU-$NH_3$ incorporates dynamic and multifactorial EFs and more comprehensive emissions sources. The determination of emission factors takes into account various parameters related to local conditions and agricultural practices. When estimating $NH_3$ emissions of synthetic fertilizer application, the model considers five types of fertilizers, as well as factors such as soil acidity, ambient temperature, fertilizer application technique and dosage, wind speed, and in-situ measurements of $NH_3$ flux. For livestock waste, $NH_3$ emissions are calculated using a mass-flow approach across four phases of manure management, considering variables such as animal rearing types, temperature and wind speed. In addition, $NH_3$ emissions from other small sources are also quantified, including agricultural soil, nitrogen-fixing crop, crop residue compost, excretion of rural populations, open biomass burning, waste disposal, gasoline vehicles, diesel vehicles, and industrial processes. The $NH_3$ emissions are allocated from provinces into 0.1 ° grids based on spatial proxies such as land cover, rural population, and other relevant indicators. Monthly emission factors shaped by meteorological conditions are used to calculate $NH_3$ emissions from fertilizer application and livestock source at a monthly level.

### 2.2 The integration of multi-source heterogeneous data

In the integration process, seven heterogeneous inventories are first normalized in terms of emission sources, species, spatial-temporal resolutions, and then integrated following a priority order to produce a standardized, highly-resolved data cube.

### 2.2.1 Source mapping

To merge inventories under a unified emission source classification system, the emission sources in the MEIC model are categorized into 88 standard sectors for mapping. The first-level category comprises 8 subcategories, namely, power, industry, residential, transportation, agriculture, solvent use, shipping, and open biomass burning. These are then further subdivided into 88 second-level sources, which take industrial classification for national economic activities for reference. For example, the industrial process sector encompasses emission sources such as the manufacturing of non-metallic mineral products,



manufacturing of chemical fibers, manufacturing of foods, smelting and pressing of ferrous metals, and more. In the initial
step of integration, the sectors from each emission inventory are mapped to the standardized two-level sources.

**2.2.2 Species mapping**

Then, non-methane volatile organic compounds (NMVOC), particulate matter (PM), and $NO_x$ in each inventory are converted
into model-ready species to support CTMs. The species mapping process is grounded in the chemical species mapping methods
in MEIC model (Li et al., 2017b; Li et al., 2014). The model supports aerosol chemical schemes such as AER05 and AER06.
$NO_x$ emissions are allocated to NO and $NO_2$ emissions based on ground observations. The step-by-step NMVOC speciation
framework developed in Li et al. (2014) is employed to generate emissions for various gas-phase chemical mechanisms
commonly used in CTMs, including CB-IV, CB05, SAPRC-07, SAPRC-99 and RADM2. The framework incorporates an
explicit assignment approach and updated profiles based on both local measurements and the SPECIATE database v.4.5. The
sources abundant with oxygenated volatile organic compounds (OVOC) are identified, and the incomplete profiles with
missing OVOC fractions are corrected. The accurate speciation mapping helps reduce uncertainties in model-ready emissions.
For inventories providing speciated VOC emissions for certain mechanisms (e.g., the YRD inventory for CB05, PRD inventory
for SAPRC-07), we directly use their emissions, or alternatively, utilize MEIC's speciation framework to generate model
species for the five chemical mechanisms.

**2.2.3 Temporal disaggregation**

The seven emission inventories are collected at different temporal resolutions (Table 1) and need to be temporally allocated to
a unified monthly scale for integration. Monthly emissions from PKU-NH$_3$, the PRD inventory, the industrial point source
inventory and MEIC can be directly used for data merge. Daily-level open biomass burning emission inventory for China is
aggregated to monthly scales through summation. For annually inventory (e.g., the YRD inventory), sector-specific monthly
profiles derived from the MEIC model are used for disaggregation (Li et al., 2017b). For instance, monthly power generation
data from the National Bureau of Statistics describe variations in monthly power emissions. Industrial production or GDP from
the National Bureau of Statistics are employed to account for monthly emission fluctuations related to industrial heating,
boilers, cement, iron and steel, and other industrial processes. Monthly emission factors calculated by the International Vehicle
Emissions model are applied to on-road vehicles. Considering the insignificant monthly variations of Automatic Identification
System data for marine shipping, the annual shipping emissions are uniformly disaggregated across the months.

**2.2.4 Spatial allocation**

The seven inventories are in different data formats, including point source and gridded formats at varying resolutions,
necessitating spatial harmonization for integration. Gridded emissions finer than 0.1 ° resolution are aggregated to 0.1 °, which
is performed in the open biomass burning inventory and the PRD inventory. For the industrial point source inventory, latitude
and longitude coordinates are employed to directly position them within grid locations. Area sources in MEIC are allocated to





grids using spatial proxies within the MEIC model (Li et al., 2017b). For instance, industrial sources are assigned to grids
based on urban population (Schneider et al., 2009). The road network (Zheng et al., 2014) serves as a proxy for disaggregating
emissions from on-road vehicles, while rural population (Schneider et al., 2009) is used as the proxy for fertilizer and livestock
sources. All the emissions are first uniformly downscaled to 1 km, and then re-gridding to 0.1 ° after the spatial-temporal
coupling process.
**2.2.5 Spatial-temporal coupling**
Finally, following the procedures outlined in Sections 2.2.1 to 2.2.4, all inventories are preprocessed to a standardized format,
encompassing 88 sectors, various species, a spatial resolution of 1 km, and a monthly temporal resolution. This preprocessing
prepares the inventories for merging, ultimately resulting in the generation of a standardized data cube.
The integration is carried out at source-by-source, species-by-species, and grid-by-grid levels, with the process guided by the
priority order of each inventory (Table 1). MEIC serves as the default inventory in our integration, offering extensive spatial
and species coverage, along with spatial proxies, temporal profiles, and NMVOC speciation methods within the model. The
remaining six emission inventories are assigned a predefined priority order. The industrial point source emission inventory for
China takes precedence over industrial emissions in MEIC, substituting proxy-based spatial allocation with precise
geographical coordinates. This extends the applicability of MEIC from a resolution greater than 0.25 ° to the kilometer scale
(Zheng et al., 2017; Zheng et al., 2021). To achieve fine-grained emission characterization in critical areas, the YRD and PRD
emission inventory enriched with localized data and advanced methods are incorporated to update emissions in these areas.
While MEIC comprehensively estimates emissions for ~800 source categories in China, there may still be omissions for certain
emission sources. The inclusion of inventories for open biomass burning and East Asian shipping helps partially fill this gap.
The PKU-$NH_3$, generated by a process-based model to provide a comprehensive understanding of China's $NH_3$ sources, is
utilized to replace all $NH_3$ emissions in other inventories. It's worth noting that the prioritization is performed city by city. For
emissions of a particular species from a specific emission sector, when multiple inventories overlap in city grids, the estimates
from the highest-priority inventory is selected as the final emissions. Through this step, the integrated inventories are developed
based on the configured output settings, such as map projection and spatial-temporal attributes.
**2.3 Evaluation of the emission inventory using WRF/CMAQ model**
We apply Weather Research and Forecasting Version 3.9 (WRFv3.9) and Community Multiscale Air Quality Version 5.2
(CMAQ5.2) as the air quality simulation systems. Two nested simulation domains with horizontal resolutions of 36 and 12
km are used (Fig. S1). The mother domain (172 × 127 cells) covers the entire China and parts of the neighboring countries,
and the nested domain (226 × 241 cells) includes the heavily polluted Eastern China. Four month (January, April, July, and
October) simulation in 2017 is carried out, with a 7-day spin-up period preceded each month. The vertical resolution in WRF
is set with 45 sigma levels ranging from the surface up to 100 hPa. Subsequently, it is collapsed into 28 layers through the
Meteorology-Chemistry Interface Processor (MCIP) before being input into CMAQ.





The configuration of WRF and CMAQ model in this study follows Cheng et al. (2019). The initial and boundary conditions
for the simulation are provided by the final reanalysis data from the National Centers for Environmental Prediction (NCEP-
FNL, https://rda.ucar.edu/datasets/ds083.2/). The schemes for shortwave radiation, longwave radiation, land surface processes,
boundary layer, cumulus parameterization, and cloud microphysics are selected as the New Goddard scheme (Chou et al.,
1998), RRTM scheme (Mlawer et al., 1997), Pleim–Xiu surface layer scheme (Xiu and Pleim, 2001), ACM2 PLB scheme
(Pleim, 2007), Kain-Fritsch scheme (Kain, 2004), and WSM6 scheme (Hong and Lim, 2006), respectively. Observational
nudging and soil nudging are employed to enhance the meteorological simulation. Regarding CMAQ model, the chemical
mechanisms for gas-phase, aqueous-phase, and aerosol are configured as CB05, the Regional Acid Deposition Model (RADM),
and AERO6, respectively. Photolysis rates are calculated online using the simulated aerosols and ozone concentrations.
Anthropogenic emissions outside China are taken from MIX inventory (Li et al., 2017b). The integrated inventory INTAC and
MEIC are used for comparison within China. Biogenic emissions are calculated using the Model of Emissions of Gases and
Aerosols from Nature version 2.1 (MEGANv2.1), while dust and lightning emissions are not considered in this study.
The performances of WRF for the meteorological parameters are evaluated against the Integrated Surface Database (ISD) from
the National Climatic Data Center (NCDC) of the National Climate Data Center (ftp://ftp.ncdc.noaa.gov/pub/data/noaa/).
Evaluation metrics include correlation coefficient (R), mean bias (MB), root mean square error (RMSE), normalized mean
bias (NMB), and normalized mean error (NME). Table S1 demonstrates good agreement between WRF model results and
ground-level observations. Similar configurations have been also validated in previous studies (Cheng et al., 2019; Cheng et
al., 2021a; Cheng et al., 2021b). CMAQ modeling results are assessed using hourly observed concentrations of air pollutants
obtained from the China National Environmental Monitoring Center (http://www.cnemc.cn/).

## 3 Results

### 3.1 China's emission characteristics in 2017

We utilized the integrated emission inventory to analyze pollutant emissions in China for the year 2017. Major air pollutant
emissions were estimated as follows: 12.3 Tg $SO_2$, 24.5 Tg $NO_x$, 141.0 Tg CO, 27.9 Tg NMVOC, 9.2 Tg $NH_3$, 11.1 Tg $PM_{10}$,
8.4 Tg $PM_{2.5}$, 1.3 Tg BC, and 2.2 Tg OC. The emission data for the eight first-level sectors can be downloaded from
https://doi.org/10.5281/zenodo.10459198 (Wu et al., 2024). The following sections will characterize emissions in detail across
sectors, fuel types, and spatial distributions.

### 3.1.1 By sectors

Table 2 displays emissions specific to the eight first-level sectors in the integrated emission inventory INTAC. For pollutants
primarily originating from fuel combustion and industrial processes (e.g., $SO_2$, $NO_x$, CO, $PM_{10}$, and $PM_{2.5}$), power, industry,
and transportation sources make a significant contribution to their emissions, ranging from 56% to 83%. Industrial sources
take a leading role in various atmospheric pollutants, contributing more than 30% for $SO_2$, $NO_x$, CO, NMVOC, $PM_{10}$, and



PM$_{2.5}$ emissions. Due to low combustion efficiency and a lack of emission control measures, residential sources exhibit a high
emission factor for products of incomplete combustion, leading to 40% of CO emissions, 48% for BC, and 73% for OC.
Solvent sources exclusively produce NMVOC emissions, constituting 33% to the overall emissions. The complexity of VOC
emission origins is evident in the diverse range of contributing sources. Agricultural sources dominate NH$_3$ emissions,
comprising an 83% share of total emissions. As described in Sect. 2.1.7, the PKU-NH$_3$ incorporates a wide variety of NH$_3$
sources, providing a more comprehensive understanding of the sectors contributing to NH$_3$ emissions. Insignificant sources
may exert large influence in specific regions or periods, such as during large wildfires or in cities with heavy traffic.
Additionally, the contribution of the supplemented open biomass burning source cannot be overlooked, especially for OC (7%)
and NMVOC (6%).
**Table 2: Anthropogenic emissions of air pollutants by sectors in the 2017 INTAC inventory for China (Units: Gg).**

| Sectors | SO$_2$ | NO$_x$ | CO | NMVOC | NH$_3$ | PM$_{10}$ | PM$_{2.5}$ | BC | OC |
|---|---|---|---|---|---|---|---|---|---|
| Power | 1822 | 3790 | 4909 | 152 | 14 | 981 | 568 | 6 | 0 |
| Industry | 6066 | 8800 | 52828 | 8824 | 249 | 5603 | 3620 | 308 | 285 |
| Residential | 2361 | 861 | 55895 | 3676 | 629 | 3516 | 3088 | 606 | 1649 |
| Transportation | 341 | 7751 | 22597 | 4123 | 619 | 533 | 493 | 257 | 95 |
| Agriculture | 0 | 0 | 0 | 0 | 7609 | 0 | 0 | 0 | 0 |
| Solvent | 0 | 0 | 0 | 9255 | 0 | 0 | 0 | 0 | 0 |
| Shipping | 1642 | 3077 | 391 | 191 | 2 | 73 | 264 | 43 | 49 |
| Open biomass burning | 21 | 215 | 4403 | 1659 | 76 | 409 | 355 | 35 | 167 |
| Total | 12253 | 24494 | 141023 | 27881 | 9198 | 11117 | 8388 | 1255 | 2245 |

Figure 2 consolidates 88 standardized emission sources into 25 categories, allowing for a more detailed analysis of sectoral
emission patterns compared to Table 2. Owing to substantial coal use in industrial and power sectors, along with sulfur-rich
ship fuels, prominent contributors to SO$_2$ emissions include power, shipping, stationary combustion, and manufacture of non-
metallic mineral products sources, accounting for 15%, 13%, 12%, and 12% respectively to total SO$_2$ emissions. This indicates
that achieving further reductions in SO$_2$ emissions will require the implementation of more energy-efficient, end-of-pipe
control measures, and adoption of low-sulfur fuels. The dominant origins of NO$_x$ emissions are from the truck, power
generation, and shipping sectors, representing 21%, 15%, and 13% of the total emissions. Both trucks and vessels extensively
use compression ignition engines, prone to generating NO$_x$ emissions under high-temperature and oxygen-rich conditions.
Implementing strict vehicle standards is crucial to effectively reduce NO$_x$ emissions from exhaust gases. Coatings, other
industrial processes, and passenger vehicle sources constitute 51% of anthropogenic NMVOC emissions. The major





contributors to primary PM$_{2.5}$ emissions include biomass fuel, the manufacture of non-metallic mineral products, and the
smelting and pressing of ferrous metals source, making up 22%, 17%, and 10% of the total emissions, respectively. It's
noteworthy that the use of biomass fuels (e.g., rice straw, firewood) for cooking or heating in rural areas results in considerable
PM$_{2.5}$ emissions, especially in provinces like Sichuan, Anhui, Shandong, and Heilongjiang.

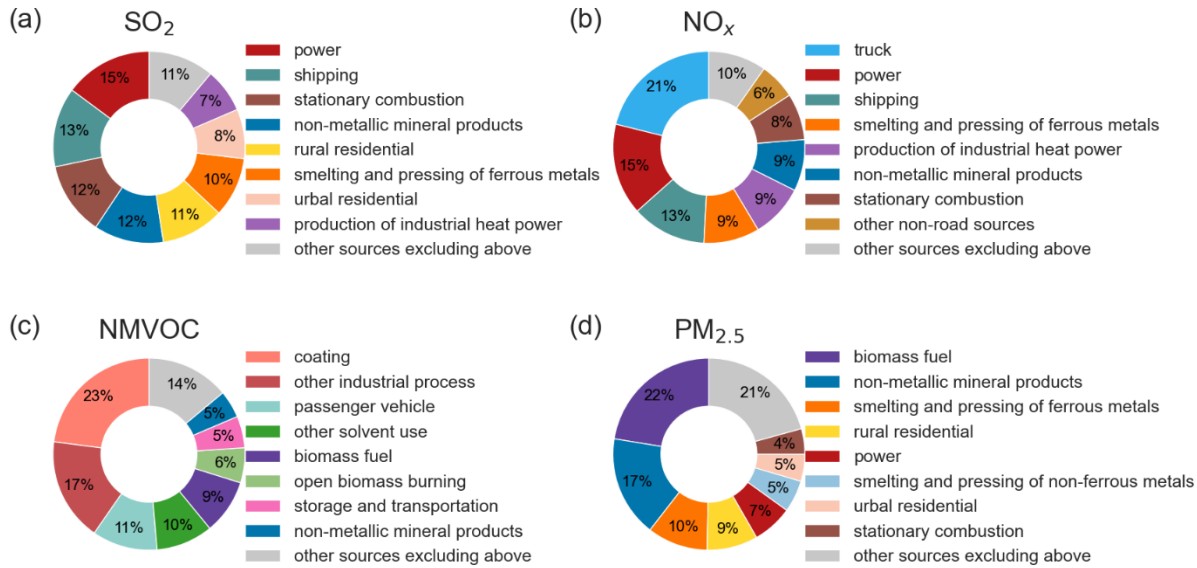

**Figure 2: Sector-specific distributions of emissions in the 2017 INTAC inventory for China.** (a), (b), (c) and (d) represent the sectoral
contributions for SO$_2$, NO$_x$, NMVOC and PM$_{2.5}$, respectively. The figure only displays the top eight contributing sources, while sources
excluding these are categorized as "other sources".

### 3.1.2 By fuel types

Figure 3 illustrates the proportions of major air pollutant emissions in 2017 for each fuel type. Fossil fuel combustion
significantly dominates the emissions of PM$_{10}$, PM$_{2.5}$, CO, BC, SO$_2$, NO$_x$, with proportion ranging from 38% to 80%. The coal
combustion accounts for 56% of SO$_2$ emissions, with power, residential activities and industrial production as the primary
emitter. Meanwhile, petroleum combustion, mainly from marine vessels, constitutes 20% of SO$_2$ emissions. For NO$_x$ emissions,
petroleum combustion contributes 48% of the total, predominantly arising from freight trucks (5.2 Tg), marine vessels (3.1
Tg), and passenger vehicles (1.0 Tg). Coal combustion processes, such as power (3.6 Tg) and industrial boiler (2.2 Tg) also
result in substantial NO$_x$ emissions (31%). The Biomass fuel source causes 53% of OC emissions. Emissions of NMVOC and
NH$_3$ are primarily associated with non-combustion processes.





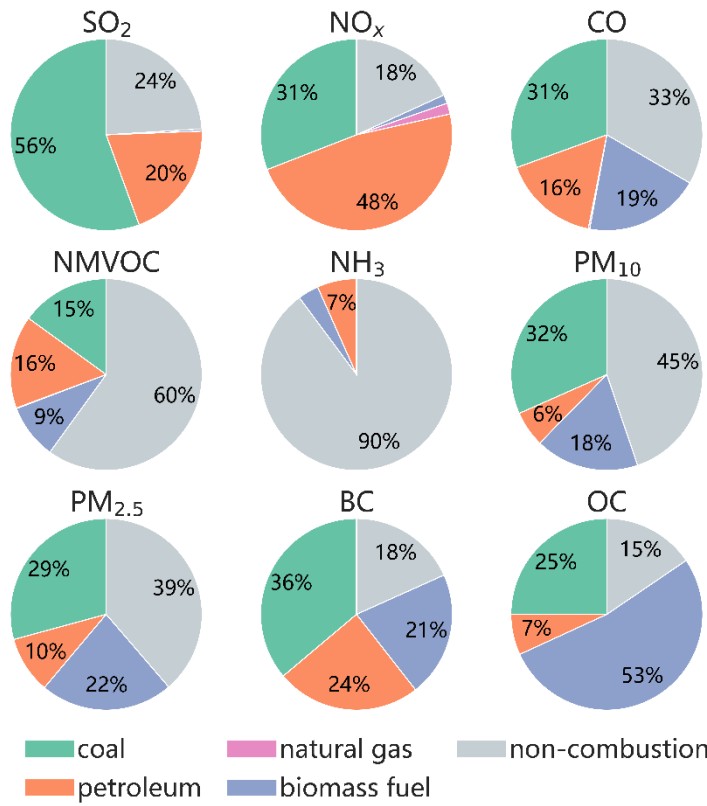

**Figure 3: Fuel-specific distributions of major air pollutant emissions in the 2017 INTAC inventory for China.**

### 3.1.3 Spatial distribution

We present the gridded emission maps of major air pollutants in Fig. 4. Emissions from anthropogenic sources in China exhibit significant spatial heterogeneity. Due to economic growth and industrial activities, air pollutant emissions are primarily concentrated in the central and eastern regions of China, especially in economically developed urban clusters such as the Beijing-Tianjin-Hebei region, the YRD, the PRD, as well as in regions like Sichuan and Chongqing. These four key areas account for 26%, 34%, 35%, 37%, 35%, 33%, 27%, 27%, and 29% of the national emissions of $SO_2$, $NO_x$, CO, NMVOC, $NH_3$, $PM_{10}$, $PM_{2.5}$, BC, and OC, respectively. Moreover, the emission maps at a fine spatial resolution of 0.1 ° × 0.1 ° depict the local variations in emission patterns, identifying numerous hotspots in small areas and showcasing distinct gradients in emissions. Table 3 shows the provincial-level emissions. The emission levels in specific provinces are determined by factors such as resource endowments, industrial structure, energy consumption, and emission control measures. Taking $SO_2$ as an example, the top five provinces are Shanxi, Shandong, Hebei, Guizhou, and Inner Mongolia, collectively accounting for 36% of the national total $SO_2$ emissions. The Guizhou Province, located in the southwest of China, is characterized by high-sulfur coal and a relatively gradual implementation of pollution control measures, which results in elevated $SO_2$ emissions. In other four provinces, large scale heavy industries have led to substantial coal consumption and correspondingly higher $SO_2$ emissions.





387    Provinces with a less industry-focused economic structure and lower energy consumption, including Tianjin, Hainan, Qinghai,

388    Beijing, and Tibet, exhibit the lowest $SO_2$ emissions, accounting for approximately 2% of the national total.

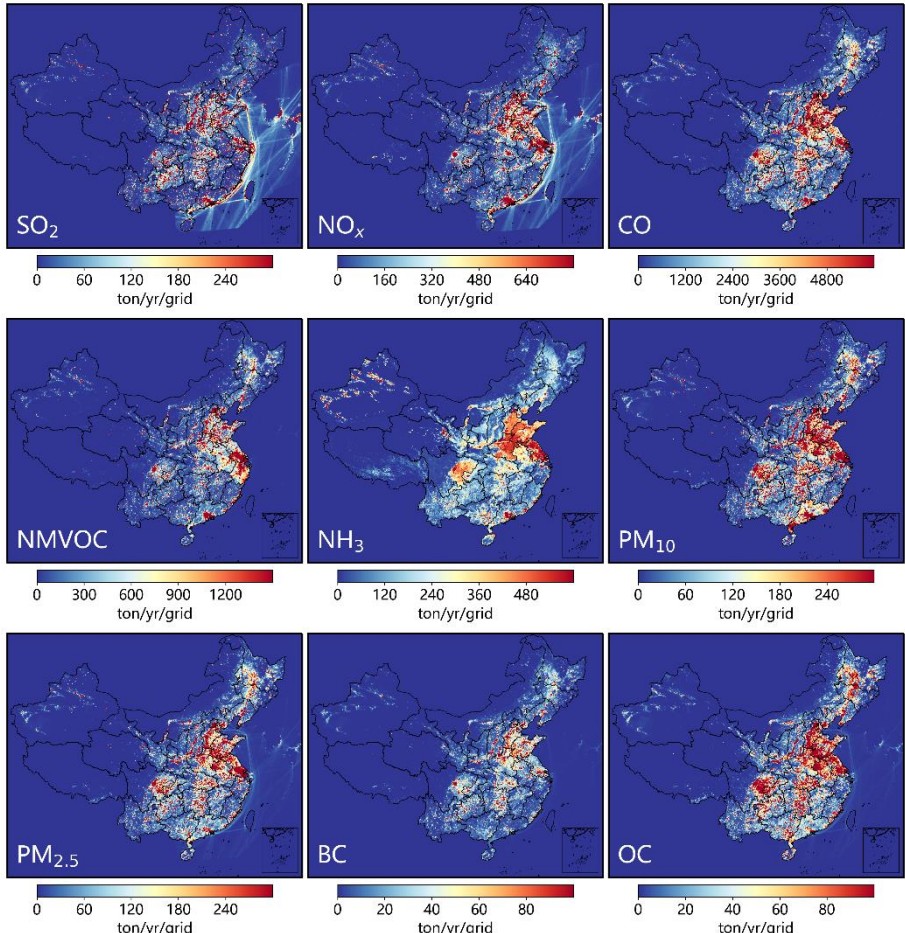

389    **Figure 4: Spatial distributions of major air pollutant emissions in the 2017 INTAC inventory for China.**



**Table 3: Anthropogenic emissions of air pollutants by provinces in the 2017 INTAC inventory for China (Units: Gg).** The shipping emission inventory in East Asia is not included.

| Sectors | SO$_2$ | NO$_x$ | CO | NMVOC | NH$_3$ | PM$_{10}$ | PM$_{2.5}$ | BC | OC |
|---|---|---|---|---|---|---|---|---|---|
| Anhui | 319 | 850 | 5968 | 1094 | 340 | 603 | 447 | 50 | 116 |
| Beijing | 27 | 232 | 1397 | 519 | 36 | 62 | 50 | 7 | 16 |
| Chongqing | 401 | 377 | 2424 | 566 | 149 | 210 | 160 | 22 | 48 |
| Fujian | 162 | 532 | 2349 | 899 | 149 | 204 | 153 | 22 | 49 |
| Gansu | 191 | 353 | 2225 | 359 | 276 | 165 | 127 | 22 | 42 |
| Guangdong | 435 | 1573 | 6912 | 1274 | 351 | 793 | 359 | 17 | 68 |
| Guangxi | 268 | 436 | 3586 | 811 | 323 | 359 | 277 | 29 | 84 |
| Guizhou | 660 | 357 | 6643 | 510 | 236 | 464 | 350 | 76 | 127 |
| Hainan | 48 | 95 | 586 | 172 | 57 | 47 | 37 | 5 | 15 |
| Hebei | 675 | 1704 | 11756 | 1681 | 522 | 717 | 532 | 88 | 126 |
| Heilongjiang | 249 | 825 | 7049 | 1426 | 378 | 501 | 407 | 65 | 158 |
| Henan | 371 | 1262 | 7979 | 1507 | 677 | 628 | 463 | 79 | 109 |
| Hubei | 519 | 706 | 6355 | 1188 | 357 | 461 | 357 | 68 | 119 |
| Hunan | 524 | 635 | 6817 | 958 | 329 | 487 | 366 | 77 | 123 |
| Inner Mongolia | 601 | 1217 | 5760 | 834 | 561 | 465 | 343 | 56 | 90 |
| Jiangsu | 395 | 1222 | 8646 | 1536 | 497 | 675 | 500 | 50 | 106 |
| Jiangxi | 181 | 451 | 3684 | 649 | 209 | 277 | 197 | 28 | 53 |
| Jilin | 238 | 655 | 3982 | 851 | 207 | 310 | 240 | 39 | 77 |
| Liaoning | 464 | 1205 | 5848 | 1322 | 268 | 437 | 328 | 54 | 88 |
| Ningxia | 228 | 329 | 767 | 179 | 79 | 92 | 63 | 7 | 9 |
| Qinghai | 44 | 107 | 599 | 130 | 131 | 60 | 45 | 5 | 8 |
| Shaanxi | 338 | 551 | 3789 | 824 | 273 | 297 | 223 | 39 | 69 |
| Shandong | 957 | 2144 | 11494 | 2859 | 694 | 907 | 684 | 105 | 152 |
| Shanghai | 116 | 471 | 1133 | 344 | 29 | 106 | 87 | 16 | 6 |
| Shanxi | 989 | 968 | 6030 | 759 | 199 | 561 | 419 | 64 | 82 |
| Sichuan | 384 | 781 | 6375 | 1485 | 644 | 468 | 374 | 56 | 143 |
| Tianjin | 91 | 335 | 1437 | 575 | 33 | 82 | 62 | 9 | 12 |
| Xinjiang | 260 | 610 | 2645 | 635 | 515 | 220 | 160 | 23 | 32 |
| Xizang | 1 | 52 | 150 | 46 | 149 | 15 | 12 | 2 | 5 |
| Yunnan | 335 | 437 | 3831 | 579 | 397 | 305 | 232 | 38 | 76 |
| Zhejiang | 297 | 672 | 3016 | 1348 | 118 | 274 | 197 | 23 | 22 |



## 3.2 Improved accuracy of China's anthropogenic emissions by HEIC

### 3.2.1 Comparison of emission magnitudes with MEIC across sectors and regions

The INTAC inventory improves the representation of anthropogenic air pollutant emissions by incorporating a large number of industrial point sources, integrating high-resolution regional inventories, and supplementing missing emission sources in MEIC. Remarkable differences between INTAC and MEIC are illustrated in Fig. 5 across regions and sectors. Compared to MEIC, the INTAC inventory shows higher levels of 16.7%, 11.5%, 10.8%, 11.0%, and 9.1% for $SO_2$, $NO_x$, $PM_{10}$, $PM_{2.5}$, and OC, respectively. However, it indicates lower levels of 6.3% and 10.6% for NMVOC and $NH_3$. CO and BC emissions exhibit good agreement between the two inventories, with differences lower than 3.9%. In comparison to MEIC, the supplementary emission sources in INTAC—specifically, open biomass burning and marine shipping—account for the majority of increased emissions, contributing 95%, 89%, and 74% for $SO_2$, CO, and $PM_{2.5}$, respectively. Additionally, the incorporation of PKU-$NH_3$ in INTAC leads to a 21% decrease in $NH_3$ emissions from agricultural sources, while $NH_3$ emissions from residential sources and transportation increase by 99% and 13.1 times, respectively. Such difference in agricultural sources is mainly caused by the estimates of synthetic fertilizer (Kang et al., 2016), particularly concerning the treatment of fertilizer types and corresponding emission factors.

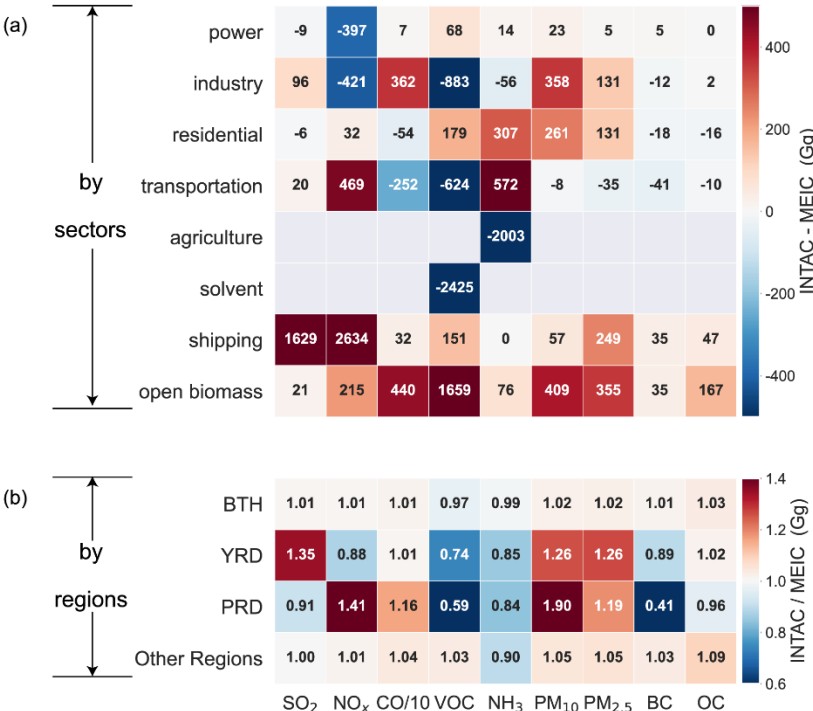

**Figure 5: Inter-comparisons of emission estimates between the INTAC inventory and MEIC.** (a) shows the difference by sectors, and (b) presents the ratio of emissions in INTAC to those in MEIC. BTH is Beijing-Tianjin-Hebei.





Many discrepancies between MEIC and INTAC arise from the integration of regional emission inventories. As presented in
Fig. 5b, notable disparities are observed in the YRD and PRD region. Estimates for NO$_x$ emissions in the YRD region are
approximately 88% of those derived from the MEIC model. This highlights an enhanced precision attributable to reliable
assessments of denitrification efficiency in power plants and the measured NO$_x$ emission factors for both power plants and
boilers within the integrated YRD inventory, as supported by previous research studies (Zhao et al., 2018). INTAC's estimates
for NMVOC emissions from the YRD region are 26% lower than estimates in MEIC. The overestimation in MEIC mainly
results from the uncertainties of solvent use source, particularly coating and printing and dyeing processes. The integrated
YRD emission inventory employs more accurate calculation parameters, such as statistical data from local city yearbooks,
industry association reports, and apparent consumption of solvents. Furthermore, the speciation profiles of NMVOC are
localized and corrected based on the literature research and measurements. In the PRD region, The NO$_x$ emissions from INTAC
are 41% higher than MEIC estimates, with non-road sources and non-metallic mineral products contributing 45% and 40% to
this difference, respectively. The PRD inventory employs a detailed calculation approach for shipping emissions based on AIS
data, in contrast to the simplified approach for inland waterway sources in MEIC. The NO$_x$ emissions from industrial processes
of brick and flat glass manufacturing are not considered in MEIC, which is a deficiency that is addressed in the integrated PRD
inventory. INTAC's NMVOC emissions are approximately 59% of those from MEIC. The disparity is particularly notable in
industrial and solvent use sources, contributing 49% and 35%, respectively, to the observed difference. In INTAC, nearly half
of the VOC emission factors for industrial solvent sources are based on local measurements, and a preference for raw material-
based calculations over product-based ones reduces uncertainty in the estimation. For significant VOC-emitting sources like
cleaning solvents, MEIC employs an emission factor of 1000 g/kg, whereas the PRD inventory uses 850 g/kg. In the case of
oil refineries, the emission factors are 2.76 g/kg for MEIC and 1.82 g/kg for the PRD inventory.
**3.2.2 Impact of point source contributions**
The most accurate method for obtaining emissions at finer-scale grids relies on spatial allocation based on precise geographical
coordinates. In MEIC, the majority of emission sources are represented as area sources and distributed onto grids using spatial
proxies such as urban population, except for power plants. In contrast, the increased proportion of industrial point source
emissions in INTAC significantly constrains the uncertainties associated with spatial proxies. Figure 6 shows the inter-
comparisons of percentage of point, on-road, and area source emissions between the INTAC and MEIC. Air pollutants,
especially those dominated by industrial combustion sources like SO$_2$, NO$_x$, PM$_{10}$, and PM$_{2.5}$, exhibit a significantly higher
proportion of point source emissions within INTAC compared to MEIC. In MEIC, the proportion of point source emissions
for SO$_2$, PM$_{10}$, NO$_x$, and PM$_{2.5}$ is 17%, 9%, 19%, and 7%, respectively. However, in the INTAC inventory, these percentages
substantially increase to 66%, 54%, 52%, and 48%, respectively, indicating a more accurate representation of spatial patterns.
For other species with emissions mainly from area sources, such as residential and transportation, there are limited
improvements in the proportion of point source emissions in INTAC.





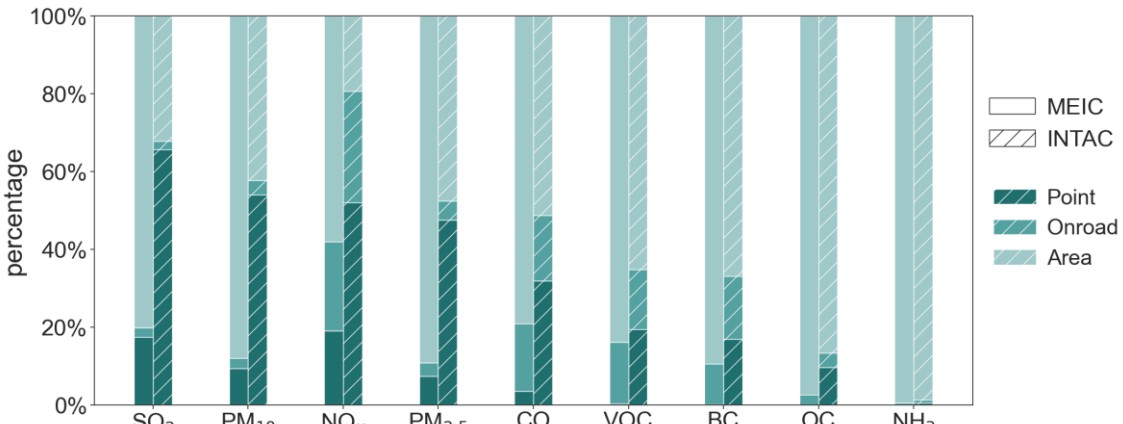

**Figure 6: Inter-comparisons of percentage of point, on-road, and area source emissions between the INTAC inventory and MEIC.**

441 To further assess the impact arising from point sources, Figure 7 takes $SO_2$ and YRD region as an example to compare the

442 spatial emission patterns between INTAC and MEIC. Figures 7c–e reveal that MEIC tends to overestimate emissions in urban

443 centers and underestimate emissions in rural areas compared to INTAC. The reliance on urban population as a spatial allocation

444 proxy in MEIC becomes unrealistic due to the relocation of factories from city centers amid economic development and rapid

445 urbanization. In Fig. 7f, the grid population is ranked from high to low to analyze the changes in cumulative $SO_2$ emissions at

446 different spatial resolutions. The findings indicate that when MEIC reaches a 50% cumulative $SO_2$ emission percentage in

447 densely populated areas at 0.05 ° resolution, INTAC only accounts for 17% of emissions. This highlights that, at a fine grid

448 scale, MEIC allocates more emissions to densely populated urban areas, while INTAC allocates a larger proportion to suburban

449 and rural areas. This reallocation in INTAC aligns better with the actual emission spatial patterns, thereby mitigating biases

450 induced by misallocation based on urban population distributions within MEIC. However, as the grid size gradually increased

451 to 1.0 °, the cumulative emissions in INTAC similarly approach 50%, demonstrating comparable performance to MEIC. This

452 can be attributed to the fact that in larger grid sizes, urban and suburban areas are often encompassed by the same grid, leading

453 to a diminishing spatial accuracy improvement caused by substituting spatial proxies with point sources. Figure 7g further

454 presents the correlation between INTAC emissions and various spatial proxies. At a resolution of 1.0 °, the correlation

455 coefficients between emission distributions and factors (i.e., road networks, nighttime lights, total population, urban population,

456 and rural population) fall within the range of 0.55 to 0.79. Nevertheless, at a resolution of 0.05 °, the correlation coefficients

457 range from 0.05 to 0.13. This indicates that at higher spatial resolutions, INTAC substantially reduces bias introduced by

458 spatial proxies in MEIC.



**Figure 7: Spatial pattern analysis of emissions in the INTAC inventory, using SO₂ emissions as an example.** (a) and (b) display the spatial distributions of SO₂ emissions in MEIC and INTAC, respectively. MEIC emissions have been downscaled from 0.25 degrees to 0.1 degrees for comparison. To compare MEIC and INTAC in details, a zoom-in is applied to the Yangtze River Delta region. (c), (d), and (e) show spatial distributions of SO₂ emissions in MEIC, INTAC and their difference. Circles in (e) represent the center of a city. (f) compares cumulative SO₂ emissions in the INTAC inventory with those in MEIC at different spatial resolutions, with the accumulated calculations performed in descending order of grid population. (g) shows correlation between SO₂ emissions in the INTAC inventory and multiple spatial proxies at different grid sizes.





**3.3 Improvements on air quality modelling by HEIC**
**3.3.1 Overall performance in key regions**
We conduct simulations using the WRF-CMAQ model driven by INTAC and MEIC separately to evaluate the improvements
in modeled air pollutant concentrations. Table 4 evaluates the simulated emissions in 74 major cities against in-situ
observations. The INTAC demonstrates an improved agreement between modeled concentrations and ground-level
observations, which benefits from the integrated high resolution inventories. Compared to MEIC, INTAC leads to a decline in
the mean bias of simulated major pollutant concentrations by 2–14 μg/m³, a reduction in the root mean square error by 4–19
μg/m³, and a decrease in the normalized mean error by 4–71%. This finding indicates that INTAC produces a more accurate
characterization of emissions in China overall. Furthermore, given that atmospheric pollution monitoring stations are mainly
located in urban areas in China, the observed differences suggest that the INTAC can mitigate the overestimation of major
pollutant concentrations in urban centers. As discussed in Sect. 3.2.2, MEIC overestimates emissions in urban areas and
underestimates them in rural and suburban areas, consequently introducing uncertainties into air quality modeling. The
enhanced accuracy in spatial distributions in INTAC significantly contributes to enhancing the overall accuracy of
concentration modeling.
**Table 4: The discrepancies between simulated SO₂, NO₂ and PM₂.₅ concentrations and observed values for 74 major cities at a**
**resolution of 12 km, using MEIC and INTAC as emission inputs.** The statistical metrics used for comparison include correlation
coefficient (R), mean bias (MB), and root mean square error (RMSE). The bold font represents the difference of modeling performance
between INTAC and MEIC.

| Pollutants | Inventory | MB ($\mu g/m^3$) | RMSE ($\mu g/m^3$) | NME (%) |
|---|---|---|---|---|
| SO₂ | INTAC | 11 | 30 | 92 |
| | MEIC | 25 | 49 | 163 |
| | **Difference** | **-14** | **-19** | **-71** |
| NO₂ | INTAC | 7 | 22 | 43 |
| | MEIC | 18 | 31 | 60 |
| | **Difference** | **-11** | **-9** | **-17** |
| PM₂.₅ | INTAC | 6 | 35 | 46 |
| | MEIC | 8 | 39 | 50 |
| | **Difference** | **-2** | **-4** | **-4** |

Figure 8 further compares the overall simulation performance between INTAC and MEIC in three key regions (BTH, YRD,
and PRD). Regarding PM₂.₅ and its precursors, MEIC shows a considerable mean bias of up to 36 μg/m³ and a root mean
square error of up to 59 μg/m³ in key regions. In contrast, INTAC demonstrates values of 15 μg/m³ for MB and 40 μg/m³ for
RMSE. The correlation coefficients between simulated and observed concentrations of the three air pollutants are generally




lower in MEIC compared to those in INTAC. The modeling performance driven by INTAC, particularly for short-lived
pollutants, experiences significant improvement due to their strong correlation with spatial distributions of emission sources.
Nonetheless, discrepancies between modeled and observed surface concentrations still exist because of uncertainties from
meteorological, physical, and chemical processes within chemical transport models. Moreover, emission sources such as
residential, transportation, agriculture in INTAC are treated as nonpoint sources, and their allocation to grids using spatial
proxies can introduce biases to air quality modeling. It is noteworthy that simulated ammonium concentrations by INTAC
agree better with ground measurements than MEIC (Table S2). While $NH_4^+$ concentrations are influenced by secondary
chemical reactions, the improved model performance still reflects the benefits from the integration of PKU-NH₃.

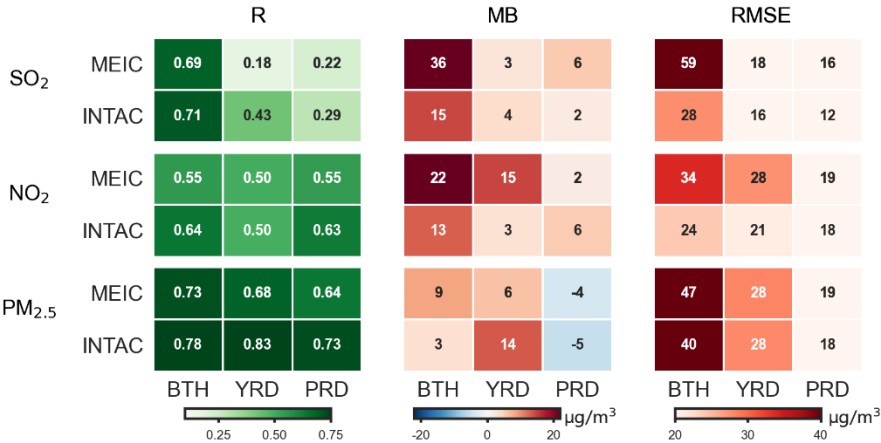

**Figure 8: The Comparison of modeling performance across key regions (i.e., BTH, YRD, PRD) when using MEIC and INTAC as emission inputs, respectively.** The statistical metrics used for comparison include correlation coefficient (R), mean bias (MB), and root mean square error (RMSE). The regions under comparison comprise the Beijing-Tianjin-Hebei (BTH), Yangtze River Delta (YRD), and Pearl River Delta (PRD).

**3.3.2 Improvements across different spatial resolutions**

For a more in-depth assessment for improved spatial patterns in INTAC, Figure 9 categorizes grid cells into different bins
based on their urban population and calculates the ratio of simulated pollutant concentrations to ground observations for both
INTAC and MEIC in each category. The results demonstrate that as urban population increases, the enhanced model
performance of INTAC over MEIC for SO₂, NO₂ and PM₂.₅ becomes more evident. Specifically, when the urban population
is less than 50,000, both INTAC and MEIC exhibit a median range of simulated-to-observed concentration ratios close to 1.
However, as the urban population exceeds 550,000, the average range for MEIC widens to 1.4–5.2, whereas it remains within
the range of 0.9–1.0 for INTAC. This indicates a significant improvement in mitigating the overestimation of emissions in
densely populated areas by INTAC. It proves that over-allocated emissions in highly populated areas due to proxy-based
methods in MEIC propagate uncertainties into chemical transport models. The incorporation of the industrial point source
emission inventory for China, along with the YRD and PRD emission inventory significantly increases point source shares in
INTAC, and thus producing better spatial representations of real-world emission distributions and smaller simulated deviations.
Discrepancies in model performance between the MEIC and INTAC are influenced by the grid sizes. Figure 10 presents the
comparison between modeled $SO_2$, $NO_2$ and $PM_{2.5}$ concentrations against ground observations for 74 major cities at resolutions
of 36 and 12 km. Increasing spatial resolution can not result in a reduction of simulation errors, particularly for MEIC. As the
horizontal resolution increases from 36 km to 12 km, the mean biases of simulated $SO_2$, $NO_2$, and $PM_{2.5}$ concentrations using
MEIC show an increase from 37% to 143%, 11% to 46%, and -3% to 15%, respectively, when compared to in-situ observations.
In contrast, the simulation results using INTAC exhibit better agreement with ground observations, with mean biases for $SO_2$,
$NO_2$, and $PM_{2.5}$ increasing from 23% to 64%, -0% to 17%, and 2% to 11%, respectively. This is due to the fact that the
deviations in finer grid cells, whether overestimated or underestimated, tend to cancel out when aggregated at a coarse spatial
resolution. The decoupling between emission spatial distributions with proxies at finer grids leads to more noticeable biases
in air quality modeling. Therefore, the findings suggest that the INTAC developed in this study can effectively constrain
uncertainties in emissions and the modeling bias, especially at fine spatial scales. The improvement will help tackle emerging
challenges in high-resolution air quality modeling in China.

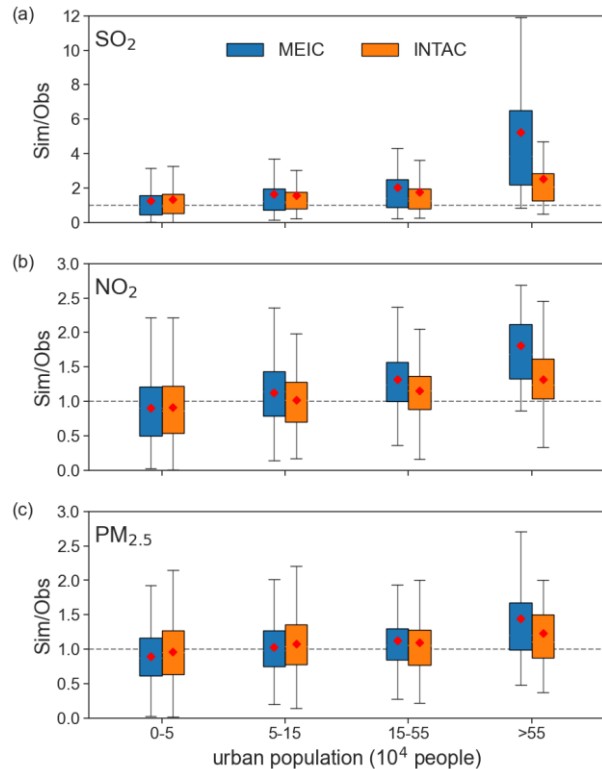

**Figure 9: Comparisons of modeling performance between INTAC and MEIC in different ranges of urban population.** The 12 km
grids are categorized to different bins according to the urban population residing within each grid. The ratio of simulated pollutant
concentrations (Sim) to observed concentrations (Obs) for major pollutants ($SO_2$, $NO_2$, and $PM_{2.5}$) are calculated. The boxplot presents the
upper quartile, median (red dot), and lower quartile of the ratios.



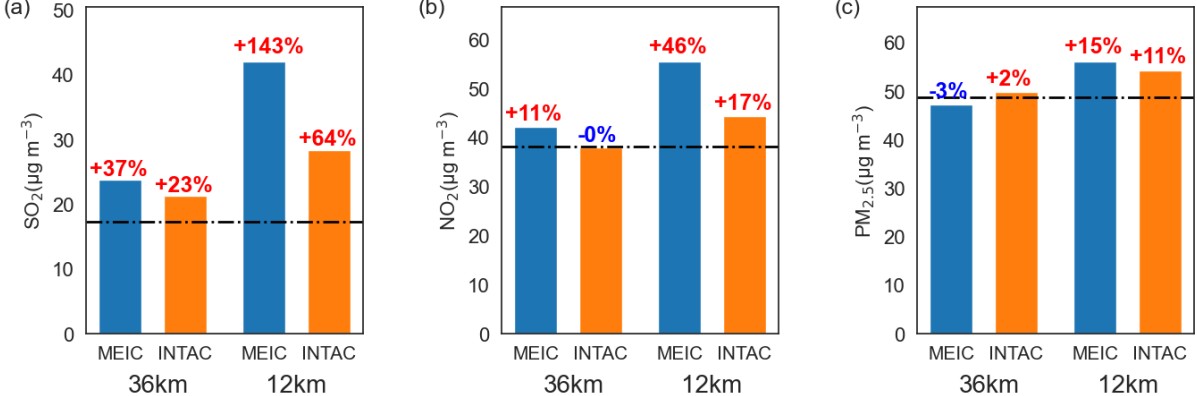

**Figure 10: The comparison of modeled air pollutant concentrations and ground observations for 74 cities at 36 and 12 km resolutions, using MEIC and INTAC as emission inputs, respectively.** The black dashed line represents the observational mean, and the annotations above the bar charts indicate the mean biases between simulated concentrations and the corresponding observed value.

## 4 Discussion

Qualitative or quantitative uncertainty assessment is a necessary element of a complete inventory for policy or scientific purposes. Approaches such as error propagation and Monte Carlo simulation are commonly used for quantitative uncertainty analysis in China's emission inventory (Zhao et al., 2017a; Zhao et al., 2011; Lu et al., 2011; Streets et al., 2003). However, this study uses an integrated method rather than a unified framework to compile a comprehensive high resolution emission inventory for China. Collecting only emission quantities from the seven inventories without detailed calculation parameters makes it challenging to assess the overall uncertainties of INTAC here. We have summarized the estimated uncertainty range for components of INTAC in Table 5, where such information is available. Although the uncertainties might be reported for a year other than 2017, they still provide a rough representation of the uncertainty range in major air pollutant emission estimates within INTAC. Species such as $SO_2$ and $NO_x$ exhibit relatively low uncertainties, benefiting from well-established estimates for large-scale combustion sources. The considerable uncertainties observed in BC and OC emissions may be attributed to inaccuracies in the emission factors of the residential sector. Further details regarding the uncertainties of each component inventory can be found in corresponding literature (Kang et al., 2016; Liu et al., 2016b; Yin et al., 2019; Huang et al., 2021; An et al., 2021; Zhao et al., 2011).

The uncertainties of INTAC also arise from the integrated process: (1) The emission source categories are based on the MEIC model, and sectors in other inventories need to be mapped to the 88 standard sectors first. Due to limited foundational information for an aggregated sector's disaggregation, this process may introduce bias for those who initially provide coarser source categories. For example, YRD only offers one aggregated sector for power, which needs to be broken down into four subsectors (i.e., production of power, supply of power, production of industrial heat power and production of residential heat power). We use the energy consumption for corresponding sectors from the statistical yearbook as a reference basis for this allocation, which is a relatively reliable method despite potential deviations. (2) To generate speciated VOC species, sectoral





NMVOC emissions in each inventory need to be matched to corresponding source profiles from the MEIC model.
Discrepancies in emission source mapping can impact the outcomes, which will be overcome by gathering more detailed
sectoral information for each inventory or directly collecting speciated species in future studies. (3) The INTAC is made
publicly available at a monthly scale, given that the majority of its components are gathered on a monthly or annual scale. The
temporal disaggregation to finer resolutions for modeling is achieved using empirically selected weighting factors in the MEIC
model. However, it is noteworthy that the parameters employed for allocating emissions to daily or hourly scales remain fixed
and do not vary over time or region, introducing additional uncertainties. In the future, we plan to incorporate more advanced
data or method (e.g., real-time emission measurements) to enhance temporal accuracy at finer scales, as indicated in the
previous work for the power sector (Wu et al., 2022). (4) The border issue is inevitable when emissions for the same species
in two adjacent cities are derived from different inventories. A typical example is the cities located at the boundary of the YRD
or the PRD regions. In the INTAC, we downscale all emissions to 1 km before spatial-temporal coupling process, thereby
mitigating this uncertainty to some extent.
**Table 5: Uncertainties in the inventory components of INTAC, contingent upon the availability of such information (Unit: %).**

| Emission inventory | Reporting year | SO₂ | NOₓ | CO | NMVOC | NH₃ | PM₁₀ | PM₂.₅ | BC | OC | References |
|---|---|---|---|---|---|---|---|---|---|---|---|
| PKU-NH₃ | 2012 | | | | | -26–25 | | | | | (Kang et al., 2016) |
| The shipping emission inventory for East Asia | 2013 | ±4 | ±4 | ±5 | ±4 | | | ±4 | ±4 | ±4 | (Liu et al., 2016b) |
| The open biomass burning emission inventory for China | 2003–2017 | -67–67 | -78–98 | -54–56 | | -44–89 | -74–84 | -65–65 | -75–100 | -74–81 | (Yin et al., 2019) |
| The PRD emission inventory | 2017 | -17–20 | -25–28 | -30–39 | -34–50 | -50–86 | -45–60 | -43–62 | -53–116 | -54–160 | (Huang et al., 2021) |
| The YRD air pollutant emission inventory | 2017 | -29–36 | -28–33 | -42–75 | -44–68 | -58–117 | -36–62 | -30–46 | | | (An et al., 2021) |
|  | 2005 | -14–13 | -13–37 | | | | -14–45 | -17–54 | -25–136 | -40–121 | (Zhao et al., 2011) |

The INTAC for 2017 is subject to some limitations: (1) The integrated method yields emissions data across various sectors
from different datasets for the same city and species, or emissions in different species from different datasets for the same city
and sector. The utilization of species ratios requires careful consideration in these cases. (2) Limited resources present a





substantial challenge in gathering emission inventories over extended time series from diverse research institutions within the
scope of this study. The shipping inventory, YRD inventory, and PRD inventory are only accessible for 2017. Consequently,
we exclusively present the INTAC for the year 2017, with the possibility of extension to other years in subsequent research.
**5 Data Availability**
Data described in this manuscript can be accessed at Zenodo under https://doi.org/10.5281/zenodo.10459198 (Wu et al., 2024).
**6 Concluding remarks**
Compiling a comprehensive bottom-up emission inventory for China that encompasses both extensive coverage and high
resolution poses a formidable challenge. In this work, we construct a 0.1 °resolution integrated inventory for 2017 through the
fusion of multi-source emission inventories. An integration model has been developed to couple heterogeneous emission
datasets, aimed at generating a standardized data cube with consistent sectors, species and spatial-temporal resolution. The
model effectively achieves the generation of a high-resolution, large-scale emission product through source mapping, species
mapping, temporal disaggregation, spatial allocation and spatial-temporal coupling. Six representative emission inventories
focusing on national and regional scales, as well as key species and sources in China are merged with MEIC, including a point-
source-based industrial inventory, two localized inventories for critical regions, two supplementary inventories covering
sources omitted in MEIC, and an $NH_3$ emission inventory. This integration capitalizes on the strengths of each inventory,
resulting in an improved depiction of emission totals and spatial distribution patterns for China.
We find that the integrated inventory provides a more comprehensive depiction of China's anthropogenic emissions. The total
emissions of $SO_2$, $NO_x$, CO, NMVOC, $NH_3$, $PM_{10}$, $PM_{2.5}$, BC, and OC in 2017 are 12.3, 24.5, 141.0, 27.9, 9.2, 11.1, 8.4, 1.3
and 2.2 Tg, respectively. Industrial production serves as the main source of various atmospheric pollutants like $SO_2$, $NO_x$, CO,
$PM_{10}$, and $PM_{2.5}$. Residential sources contribute over 40% to CO, BC and OC emissions. Apart from agricultural sources,
which account for 83% of $NH_3$ emissions, it is crucial to acknowledge the contributions from various minor emission sources.
This study emphasizes the significance of shipping emissions, particularly in contributing to $SO_2$ (13%) and $NO_x$ (13%). Fossil
fuel combustion dominates the emissions of $PM_{10}$, $PM_{2.5}$, CO, BC, $SO_2$, and $NO_x$, ranging from 38% to 80%. Compared to
MEIC, INTAC has greatly improved emission estimates in China. For instance, by integrating the YRD inventory, more
accurate quantification of $NO_x$ emission factors and denitrification efficiency results in a 12% reduction in $NO_x$ emissions in
INTAC. The incorporation of numerous point sources in INTAC has notably addressed MEIC's tendency to overestimate
emissions in urban centers, particularly at higher spatial resolutions. The reduction in emission biases leads to a decrease of
uncertainties propagated into the CTMs. In comparison to MEIC, INTAC has exhibited a mean bias reduction in simulated
concentrations of major pollutants against ground observations across 74 cities, ranging from 2–14 µg/m³. The improvement
in model performance achieved by INTAC is particularly noticeable at finer spatial resolutions.



Our study offers an efficient framework for creating highly-resolved emission inventory on a large scale. This framework
integrates advantages from previous studies conducted by multiple research organizations and holds the potential to aid
policymakers in making well-informed decisions for improving air quality. In the future, we expect the incorporation of a
growing number of emission datasets to offer a more reliable representation of emissions in China.
**Supplement**
The supplement related to this article has one figure (i.e., Figure S1) and two table (i.e., Table S1, S2).
**Author contributions**
Nana Wu, Guannan Geng, and Qiang Zhang designed the study. Nana Wu developed the INTAC emission inventory and
conducted chemical transport modeling. Junyu Zheng, Yu Song, Huan Liu, Yu Zhao, Ying Zhou and Qinren Shi provided the
emission inventories for the integration. Ruochong Xu helped with the data analysis. Shigan Liu compiled the chemical
transport model. Xiaodong Liu contributed to the design of computer programmes for the integration model. The manuscript
was written by Nana Wu and Guannan Geng, and it was revised and discussed by all coauthors.
**Competing interests**
The authors declare that they have no conflict of interest.
**Acknowledgements**
This work was supported by the National Natural Science Foundation of China (Grant No. 92044303), the National Key R&D
program of China (Grant No. 2022YFC3700605), and the Major Project of High Resolution Earth Observation System (Grant
No. 30-Y60B01-9003-22/23). We thank Zhijiong Huang, Junchi Wang, Mingxu Liu, Wenling Liao, Chen Gu for their
contributions to the handling and transfer of the emission inventories for the integration.

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
