# Peer review of "Development of a High-Resolution Integrated Emission Inventory of Air Pollutants for China"

_Earth System Science Data, 2024_

## Author Comment (AC1)

•**General comment**: This manuscript is well written, and most of the data presentation is clear and comprehensive. The amount of work is substantial given that multiple emissions inventories in different structures are integrated. And this work will be very useful for future studies. This manuscript can be even better if the specific comments listed below can be addressed.

Response: We thank Referee #1 for the encouragement and the constructive comments to improve our manuscript. All the comments have been addressed in the revised manuscript.

•**Specific comment 1**: Fig. 1: how is the priority determined? Please provide more details on how the decision was made and why one inventory can be more believable than another?

Response: We thank the reviewer for the valuable comments. The priority order of these emission inventories is determined based on the comprehensive comparison and evaluation between different emission inventories. Given MEIC's extensive coverage across species, sectors, and spatial domains, it is well-suited to serve as the default inventory in this study and can supplement the missing data in other inventories. In addition to MEIC, the remaining six inventories can be categorized into three types in sequence — point-source inventories (i.e., the industrial point source emission inventory for China), regional inventories (i.e., the YRD emission inventory, the PRD emission inventory) and process-based inventories (i.e., the open biomass burning emission inventory for China, the shipping emission inventory for East Asia, and the PKU-NH$_3$). **(1)** The point-source-based inventory can directly correct the spatial misallocation of industrial emissions by MEIC at fine scales (Zheng et al., 2021; Zheng et al., 2017). As per Zheng et al. (2021), the industrial point source emission inventory for China reduces modeling bias from 27% to 5% for PM$_{2.5}$ at a horizontal resolution of 4 km. **(2)** Compared to the point-source-based inventories, the regional inventories further enhance local investigations of individual emission sources and simultaneously refine estimation methods for mobile and area sources (Gu et al., 2023; Huang et al., 2021; Sha et al., 2021; Zhao et al., 2018; Zhou et al., 2017). Taking the YRD region as an example, the proportion of point source SO$_2$ emissions in the regional inventory is 79%, whereas in the MEIC, after integration with national industrial point-source inventories, it is 64%. Regional inventories have been shown to exhibit better agreement with measurements compared to MEIC (Zhao et al., 2018; Zhou et al., 2017). **(3)** Process-based inventories typically adopt advanced methods to improve the

characterization for emission processes and parameters specific to particular sectors or species, thereby providing emission totals and distributions that are more in line with reality (Huang et al., 2012a; Huang et al., 2012b; Kang et al., 2016; Liu et al., 2016; Liu et al., 2019; Liu et al., 2015; Song et al., 2009; Yin et al., 2019). For example, the PKU-NH$_3$ incorporates dynamic and multifactorial emission factors, taking various parameters related to meteorological factors, soil properties, and agricultural practices into account. The process-based model has been shown to closely match top-down NH$_3$ inversions (Paulot et al., 2014), and the PKU-NH$_3$ is utilized to replace the NH$_3$ emissions from corresponding sectors in all other inventories.

We now rewrite the paragraph before Sect. 2.1.1 in the revised manuscript to add more explanations: "Table 1 lists the essential details about the seven inventories and priority order utilized for integration. Given MEIC's extensive coverage across species, sectors, and spatial domains, it functions as the default inventory in our integration, supplementing the missing data in other inventories. The remaining six inventories can be categorized into three types in sequence: point-source-based inventory (ranked sixth), regional inventories (ranked fifth and fourth), and process-based inventories (ranked third to first). The point-source-based inventory can directly correct the spatial misallocation of industrial emissions by MEIC at fine scales (Zheng et al., 2021; Zheng et al., 2017). The regional inventories further enhance local investigations of individual emission sources and simultaneously refine estimation methods for mobile and area sources (Gu et al., 2023; Huang et al., 2021; Sha et al., 2021; Zhao et al., 2018; Zhou et al., 2017). Process-based inventories typically adopt advanced methods to improve the characterization for emission processes and parameters specific to particular sectors or species, thereby providing emission totals and distributions that are more in line with reality (Huang et al., 2012a; Huang et al., 2012b; Kang et al., 2016; Liu et al., 2016; Liu et al., 2019; Liu et al., 2015; Song et al., 2009; Yin et al., 2019)."

•**Specific comment 2**: Line 126: please provide more details on how the industrial point sources were incorporated into INTAC. How is mass conserved when doing such point sources incorporation?

Response: The industrial point source emission inventory was developed by the MEIC team, which has considered its coupling with MEIC during the development process (Zheng et al., 2021). They used the activity data in MEIC as a total constraint to ensure mass conservation in this process. Specifically, the activity data of each industrial source in the point source

inventory is scaled to match the national totals of that in MEIC. Then, the point sources are mapped to standard source classification to replace corresponding emission sources in MEIC. This adjustment is necessary because the MEIC relies on provincial statistics, which offer a more accurate representation. While the activity data from all point sources tends to underestimate emissions, potentially omitting small factories. More details can be found in Zheng et al. (2021).

We now add more details in Sect. 2.1.2 in the revised manuscript: "It is worth noting that the facility-level activity data was corrected using provincial activity data from MEIC as a total constraint to be consistent with national totals from statistics (Zheng et al., 2021)."

•**Specific comment 3**: Line 276: why the final product needs to be re-gridded to 0.1 degree even though that you were able to downscale to 1km?

Response: Regarding the seven inventories in this study, two of them can accurately pinpoint the specific geographic locations of emission sources. Specifically, the industrial point source emission inventory for China employs the latitude and longitude coordinates of industrial facilities to locate emission within 1-km grids, which ensures spatial accuracy as demonstrated in Zheng et al. (2021). The open biomass burning emission inventory for China uses fire count locations obtained from satellite observations for the spatial allocation of emissions (Huang et al., 2012a). However, due to the lack of accurate geophysical locations for all emission sources in the other five inventories, we have to rely on numerous spatial proxies (e.g., population) to disaggregate emissions into 1-km grids. As mentioned in the introduction, this approach may introduce biases into emission spatial distributions and chemical transport models. To ensure the highest level of accuracy, we re-grid the final product to 0.1 degrees.

It's important to mention that uncertainties may arise at city borders if emissions from adjacent cities come from different inventories during the integration process. To mitigate biases introduced by border issues, all emissions at 0.1° resolution are first uniformly downscaled to 1 km for the spatial-temporal coupling process, and then re-gridded back to 0.1°.

We now add more explanations in Sect. 2.2.4 of the revised manuscript: "Although the industrial point source inventory and the open biomass burning inventory can accurately pinpoint the specific geographic locations of emission sources, the other five inventories rely on numerous spatial proxies to disaggregate emissions into grids, which inevitably introduce

uncertainties at very fine resolutions. Therefore, we re-grid the final product to 0.1° to ensure high level spatial accuracy."

•**Specific comment 4**: Fig. 7f: this figure is useful to demonstrate the point you want to make, but it is kind of hard to understand given its current format, caption, and text description starting line 445. For example, I wasn't sure what the percentage numbers on the figure mean and I wasn't aware that the vertical line was for 50% on the x axis. Some more detailed description need to be added either to the figure itself, in the caption, or in the text discussing the figure.

Response: We rephrase the text in the revised manuscript: "To elucidate the difference between population-based and point-source-based allocation methods in emissions mapping, we present the cumulative percentage of $SO_2$ emissions in MEIC and INTAC based on descending population orders in Fig. 7f. We use the grid groups where densely populated areas contribute 50% of $SO_2$ emissions in MEIC as an example, comparing them with the cumulative percentage in INTAC across various grid sizes. The results indicate that at a resolution of 0.05°, INTAC only accounts for 17% of the emissions, while it reaches to 48% as the grid size increases to 1.0°. This suggests that at a fine grid scale, MEIC tends to allocate more emissions to densely populated urban areas, while INTAC allocates a larger proportion to suburban and rural areas, aligning better with the real-world emission spatial patterns. This mitigation of bias through INTAC is especially notable at finer resolutions. The close cumulative percentage at 1.0° in the two inventories can be attributed to the fact that urban and suburban areas often fall within the same grid, leading to a decreasing enhancement in spatial accuracy achieved by INTAC."

As depicted in the figure below, we have adjusted the axis labels in Fig. 7f and 7g to enhance the clarity of the graph's message, while also providing additional details in the caption.

[Figure]

**Figure 7:** (f) compares cumulative percentage of SO₂ emissions in the INTAC inventory with those in MEIC across different spatial resolutions. The gridded SO₂ emissions, ranging from resolutions of 0.05° to 1.0°, are cumulated in descending order of populations. The percentage annotations in different colors indicate the level of accumulated SO₂ emissions in INTAC at various spatial resolutions when SO₂ emissions in MEIC reach 50% accumulation.

**Technique comment 1**: Line 100: the singular of species is still species.

Response: "specie" has been changed into "species".

**Technique comment 2**: Line 162: RPD to PRD.

Response: "RPD" has been changed into "PRD".

**Technique comment 3**: Line 169: this is the first occurrence of "AIS" while it is not spelled out until line 205. Please make sure every acronym is spelled out at its first occurrence.

Response: "AIS" and other acronyms have been defined the first time they are used in the text.

**Technique comment 4**: Line 383: when discussing provinces, I think it is useful to provide a province boundary map in the S.I. for readers not familiar with Chinese geography.

Response: We thank the reviewer for the valuable comments. We have now included the province boundary map in the SI (Fig. S2). And a sentence has been added in the Sect. 3.1.3

for improved reference: "Table 3 shows the provincial-level emissions (except Hong Kong, Macao, and Taiwan), and a map depicting provincial boundaries is displayed in Fig. S2."

[Figure]

**Figure S2: The location of key regions, provinces, and 74 cities in China.** The shaded area in orange from north to south represents the BTH, YRD, and PRD regions. Hong Kong, Macao, and Taiwan are excluded provinces due to unavailability of emission data. The red dots denote the locations of the 74 major cities.

**Technique comment 5**: Line 392/466: what is HEIC? I guess it is the previous name of INTAC?

Response: "HEIC" is indeed the original name of the dataset. It has been changed into "INTAC" in the revised manuscript.

**Technique comment 6**: Line 407: same as comment 3, BTH can be introduced on line 377, where it shows up for the first time.

Response: The abbreviations for BTH, YRD, PRD and others are spelled out in the text at first use.

**Technique comment 7**: Line 469: Could you please provide a map that has the locations of the 74 major cities in the SI?

Response: We have provided a map for the locations of the 74 major cities in the SI (Fig. S2).

**Technique comment 8**: Table 4, Fig. 8: providing the overall statistics is concise but some readers might want to see the raw data points in scatter plots. These can provide information such as the number of data points, the scatter distributions, etc. Some times, the overall bias can be driven by a few outliers. And the number of data points matters for the statistics you are presenting here. These can be put in the SI.

Response: We thank the reviewer for the valuable comments. As suggested, scatter plots have been included in the supplementary information (Fig. S3 – Fig. S6).

**Reference**

[revised manuscript text omitted]

---

## Author Comment (AC2)

**Referee 2#**

In this paper, Wu et al. combined several existing high-resolution emission inventories to develop a highly accurate dataset for China. This integrated approach, instead of the traditional bottom-up method relying on fundamental emission rates and factors, facilitates easier construction of large-scale and high-spatiotemporal-resolution emission inventory. The resulting integrated inventory highlights an increased proportion of point source emissions, along with enhanced accuracy in emission magnitudes and spatiotemporal patterns. The figures in the paper offer clear evidence of how the new inventory has improved the model performance. Compared to the widely-used China's emission inventory MEIC, which is applicable at resolutions lower than 0.25 degrees, this new $0.1°$ dataset is proved to be a highly valuable asset for researchers in the field of emission inventory development and air quality modeling. The paper is well-written, logically structured, and straightforward. I would recommend publication after a minor revision.

Response: We thank Referee #2 for the encouragement and insightful comments on our manuscript. Below, we provide responses to each of your points to improve our work.

• **Comment 1**: Why is the integrated emission inventory only constructed for the year 2017? Would it be extended to have a time series or more recent years in the future?

Response: The establishment of INTAC is the outcome of collaborative efforts among multiple Chinese institutions, with support from organizations like the National Natural Science Foundation of China. The data collection process posed significant challenges. We selected 2017 as our focal year based on the intersection year for each collected inventory component. Additionally, 2017 marked the conclusion of China's most stringent Air Pollution Prevention and Control Action Plan. Recognizing the importance of this dataset in both atmospheric science and policy research, we aim to extend our dataset in the future.

We have a short discussion in Sect. 4 of the manuscript: "Limited resources present a substantial challenge in gathering emission inventories over extended time series from diverse research institutions within the scope of this study. Consequently, we exclusively present the INTAC for the year 2017, with the possibility of extension to other years in subsequent research."

•**Comment 2**: Why not integrate $CO_2$ in this study? While it's not classified as an air pollutant, it's a crucial species to consider.

Response: $CO_2$ holds significant importance in emission inventories for climate research and emission mitigation policies. However, it's not included in our work due to we need to comprehensively consider the species provided by each inventory. Therefore, the INTAC only focuses on air pollutants. We aim to extend our dataset to include $CO_2$ in the future.

•**Comment 3**: In the section 2.2.1, could you provide more details about the 88 standard sectors? I think a supplementary table would be helpful. I'm also a bit confused about the sectors in the legends of Figure 2. There are sectors labeled "passenger vehicle" or "truck", but also one called "storage and transportation". Could you clarify the relationships between those vehicles and transportation?

Response: We thank the reviewer for the valuable comments. We have added a table in the supplementary information, labeled as Table S1.

The "storage and transportation" refers to storage and transportation of crude oil and natural gas, which has been revised in the legend of Figure 2. The difference between "passenger vehicle" or "truck" lie in the intended purpose and capacity. The "passenger vehicle" is classified for passenger transportation, encompassing mini passenger cars, small-duty passenger cars, medium-duty passenger cars, and heavy-duty passenger cars. The "truck" is used for freight transportation, which includes mini-duty trucks, light-duty trucks, medium-duty trucks and heavy-duty trucks, as well as low-speed freight trucks and three-wheeled vehicles. To enhance clarity, we have replaced "truck" with "freight truck" in the revised manuscript.

•**Comment 4**: In Figure 2, the legends are so close to the pies. It would be better if this is modified.

Response: The legends have been modified as suggested.

•**Comment 5**: The conclusion is a little long and should be shorten.

Response: The conclusion has been shortened as suggested.